# Ordered Subspace Clustering for Complex Non-Rigid Motion by 3D Reconstruction

**Weinan Du** , **Jinghua Li \***, **Fei Wu, Yanfeng Sun and Yongli Hu**

Beijing Key Laboratory of Multimedia and Intelligent Software Technology, Faculty of Information Technology, Beijing University of Technology, Beijing 100124, China; duweinan@emails.bjut.edu.cn (W.D.); dizhouxian201314@gmail.com (F.W.); yfsun@bjut.edu.cn (Y.S.); huyongli@bjut.edu.cn (Y.H.)

\* Correspondence: lijinghua@bjut.edu.cn

**Abstract:** As a fundamental and challenging problem, non-rigid structure-from-motion (NRSfM) has attracted a large amount of research interest. It is worth mentioning that NRSfM has been applied to dynamic scene understanding and motion segmentation. Especially, a motion segmentation approach combining NRSfM with the subspace representation has been proposed. However, the current subspace representation for non-rigid motions clustering do not take into account the inherent sequential property, which has been proved vital for sequential data clustering. Hence this paper proposes a novel framework to segment the complex and non-rigid motion via an ordered subspace representation method for the reconstructed 3D data, where the sequential property is properly formulated in the procedure of learning the affinity matrix for clustering with simultaneously recovering the 3D non-rigid motion by a monocular camera with 2D point tracks. Experiment results on three public sequential action datasets, BU-4DFE, MSR and UMPM, verify the benefits of method presented in this paper for classical complex non-rigid motion analysis and outperform state-of-the-art methods with lowest subspace clustering error (SCE) rates and highest normalized mutual information (NMI) in subspace clustering and motion segmentation fields.

**Keywords:** low rank representation; subspace clustering; non-rigid structure-from-motion

## 1. Introduction

Modeling and analysis of non-rigid motions from image sequence are challenging problems in computer vision due to complex deformable pattern and shape structure, for example, dynamic scenes, human body activities, expressive or talking faces, etc. NRSfM tries to restore 3D non-rigid features and camera movements out of 2D point tracks collected by a monocular camera, which has received increasing attentions in the related community. Generally, two classes of current popular NRSfM methods roughly are: shape basis factorization and correspondence. The shape basis methods [1–3] assume principal components of non-rigid motion and then utilizes different shape bases to give a linear representation. The correspondence approaches [4,5] aim to reconstruct 3D motions from dense points or each pixel in the image sequence, which generally need spatial constraints as regularizers. Although current NRSfM methods work well in reconstructing simple non-rigid deformations, these NRSfM methods still have problems when it comes to practical scenarios with complex non-rigid shape variations and different kinds of motions, such as human activities of sitting, walking, bending, dancing etc.

Recently, Dai et al. [6,7] adopted a simple subspace to model non-rigid 3D shapes and proposed a "prior-free" method for NRSfM problem, where there is no prior assumption about the non-rigid or camera motions. In fact, the method can be regarded as an extension of the Robust Principal Components Analysis (RPCA) method [8]. RPCA aims to recover low-rank subspaces from noisy data.

Dai's method makes the same assumption as RPCA, however, Dai's method is oriented to 3D data reconstructed from the known 2D information. Unfortunately, the method suffers from low accuracy in reconstructing complex and various non-rigid motion. Although this method has been improved by iterative shape clustering [9], which reconstructs 3D shapes and clusters the ones recurrently, however, the improved method still faces problems for dealing with the complex non-rigid motion.

Considering complex non-rigid motion in NRSfM, Zhu et al. [10] followed Dai et al. [6,7] and proposed a method named complex non-rigid motion 3D reconstruction by union of subspaces (CNRMS), which reconstructs 3D complex non-rigid motion from 2D image sequence with relative camera motions. In this method, the clustering for 3D non-rigid motion is simultaneously implemented by a union of subspaces. It is considered that the union specifics of the individual subspaces are more suitable to model the complex and various non-rigid motion. The experimental results show that the method has higher clustering accuracy and better reconstruction results compared with the method in [6].

Though introducing a subspace clustering method into the NRSfM model brings considerable improvement for both 3D reconstruction and motion segmentation, current NRSfM methods usually apply fundamental subspace clustering theory. However, the clustering research has had many successful applications in computer vision, pattern recognition, and image processing [11–13], especially spectral clustering of subspace clustering methods by affinity matrix have better performance. For example, the segmentation accuracy in [14] is three percentage points higher than RPCA, since the constraint of affinity matrix is helpful for data representation [13–15]. According to the representative clustering methods [16–20], it is interpreted that to obtain good clustering results, the intrinsic property of the data should be explored, and the feasible structure of the affinity matrix should also been considered. However, the current methods of NRSfM take no account of these factors. On one hand, the sequence property of non-rigid motion has not been considered in current methods, which is ubiquitous in motion segmentation and other applications involved in sequential data. It is proved by ordered subspace clustering (OSC) [16,17] methods which is recently proposed, that clustering using sequential or ordered properties will improve clustering accurancy significantly. On the other hand, there is no constraint for the structure of the affinity matrix in current NRSfM methods. From the success of the subspace clustering methods [17–19], which adopted structure of the affinity matrix such as block-diagonal property, it is necessary to make constraints for the structure of the affinity matrix in NRSfM clustering methods.

Kumar et al. [21] proposed a joint framework that both segmentation and reconstruction benefit each other. In the trajectory space and shape space there are multiple subspaces with better reconstruction results. While Dai et al. [22] demonstrated a different view about dense NRSfM problem by considering dense NRSfM on a Grassmann manifold.

In this paper, based on ordered subspace clustering for complex and various non-rigid motion by 3D reconstruction, a novel method is proposed, where the sequential property is properly formulated in the procedure of learning the affinity matrix for clustering with simultaneously recovering the 3D non-rigid motion from 2D point tracks. Experiments results on a few sequential datasets show the benefits of the proposed model about the complex non-rigid motion analysis and its results outperform state-of-the-art methods of motion segmentation problem. The contributions are mainly listed in detail:

- A novel framework is proposed for segmentation of complex and various non-rigid motion from 3D reconstruction using ordered subspace clustering.
- Instead of nuclear normal, a quadratic constraint is used in the self-representation model to improve the clustering performance.
- An efficient algorithm is implemented solving the complicated optimization involved in proposed framework.

This paper is organized as follows. First, related works are presented in Section 2. Section 3 gives the proposed model in detail. The solution to the optimization model above is given in Section 4.

In Section 5, the proposed method along with state-of-the-art methods are evaluated on several public datasets. Finally, we give the conclusions of this paper in Section 6.

## 2. Related Works

We briefly summarize the representative methods of NRSfM, particularly the NRSfM method in [6,10].

In recent works, Dai et al. [6] proposed a method of NRSfM which adopted a low-dimensional subspace to model the non-rigid 3D shapes, which is similar to the principle of RPCA, proposed by Cand'es et al. [8]. The RPCA approach usually represents the noisy 2D data derived from a low-dimension subspace, where the clean data is recovered by the following objective function,

$$\min_{D,E}\|D\|_* + \lambda\|E\|_l, \text{ s.t. } X = D + E, \tag{1}$$

where $D$ is the clean data of low-dimension subspace with low-rank constraint modeled by nuclear normal $\|\cdot\|_*$. $E$ denotes residual noise, and $X$ denotes corrupted data with noise. $\|E\|_l$ denotes error norm, when $l = 1$ it is the sparse error and $l = 1, 2$ it denotes group sparse error. The error penalty parameter is noted by $\lambda > 0$.

The NRSfM model proposed by Dai et al. [6] can be regarded as an extension of RPCA, and the overall model is described below,

$$\min_{X,E}\|X\|_* + \lambda\|E\|_2, \text{ s.t. } W = RX^\# + E, \tag{2}$$

where $W \in R^{2F \times N}$ is the known 2D information of N points, which also can be seen in the sequences projected from 3D points coordinates noted by $X^\# \in R^{3F \times N}$, here, the number of 3D points is $N$ and there are totally $F$ frames. $X \in R^{F \times 3N}$ is a reshape of $X^\#$. The projection matrix is denoted by $R \in R^{2F \times 3F}$, which can be pre-computed out of the 2D sequences by the methods in [6,23]. In this method, the $l_2$ norm is selected for the error $E$ under the assumption of Gaussian noise. It can be replaced by other norms as in RPCA. It is called a "prior-free" method as there is no prior assumption on non-rigid framework or camera motions.

Zhu et al. [10] argued that Dai et al.'s method [6] assumed the data sampled from a subspace and suffered the same low accuracy as RPCA, when it is applied to complex and various motion reconstruction. Zhu et al. proposed a NRSfM method by modeling complex and various non-rigid motion as subspace set inspired from subspace clustering method, the low rank representation (LRR) [14,15], and the model is described below,

$$\min_{X,Z,E}\|Z\|_* + \lambda\|E\|_l + \gamma\|X\|_*, \text{ s.t. } W = RX^\# + E, X = XZ. \tag{3}$$

where $X = XZ$ stands for subspace clustering constraint which automatically enforces structure of subspace set $X$, with $Z$, stands for a matrix of low rank coefficients. $W = RX^\# + E$ constrains the 3D reconstruction out of 2D projections, that is, $W$ to $X$. The penalty parameter for $\|X\|_*$ is $\gamma$ and the penalty parameter of $\|E\|_l$ is $\lambda$, $\lambda > 0$. It is shown that the method simultaneously reconstructs and clusters the 3D complex non-rigid motions $X$, which involved a subspace set by low-rank affinity matrix $Z$.

Recently, ordered subspace clustering (OSC) [16] and the ordered subspace clustering with block-diagonal priors (QOSC) [17] is proposed. We try to model the sequential property of sequence data in the view of subspace clustering.

## 3. The Proposed Model

In the real world, most of the motions are continuous and sequential, especially for the videos captured by a monocular camera. However, current methods about NRSfM do not utilize the sequential or ordered information embedded in the non-rigid motion data. As for this problem, the OSC [16] and

the QOSC [17] give a proper way to model the sequential property. Motivated by OSC and QOSC, we introduce a penalty term to penalize the similarity between consecutive columns of the low rank representation $Z$ from reconstructed 3D motion data $X$, thus we obtain the following NRSfM model:

$$\min_{X,Z,E} \frac{1}{2}\|X - XZ\|_F^2 + \frac{\lambda}{2}\|Z\|_* + \lambda_1\|ZS\|_{2,1} + \lambda_2\|X\|_* + \lambda_3\|E\|_1 \tag{4}$$
$$\text{s.t. } W = RX^{\#} + E,$$

where $S$ is a triangular matrix only consisting of $-1$, 1, and 0 values, with the diagonal elements being $-1$ and the second being 1,

$$\mathbf{S} = \begin{pmatrix} -1 & 0 & 0 & \cdots & 0 \\ 1 & -1 & 0 & \cdots & 0 \\ 0 & 1 & -1 & \cdots & 0 \\ \vdots & \vdots & \ddots & \ddots & \vdots \\ 0 & 0 & 0 & \ddots & -1 \\ 0 & 0 & 0 & \cdots & 1 \end{pmatrix}_{n \times (n-1)},$$

which can make consecutive columns of $Z$ alike. Thus $\|ZS\|_{2,1}$ seeks to preserve the sequential property of $Z$ for penalty , which is determined by the inherent sequential property of the the motion data $X$. Additionally, the norm $\|\cdot\|_{2,1}$ is used to maintain the sparsity. It is also denoted that the rigid constraint of $X = XZ$, the low rank self-representation, is regarded as a reconstruction error in the objective function.

To further obtain good clustering performance, we tried to improve the current methods by modeling the structure of the affinity matrix of the LRR representation $Z$. In subspace clustering, the ideal result of representation coefficients of inter-subspace items are all zeros and only items from the same subspace are non-zeros. Thus its certain permutation and affinity matrix drawn from different subspaces are block-diagonal. Therefore the clustering performance is improved by utilizing the block-diagonal prior. One reprehensive method is the subspace segmentation via quadratic programming (SSQP) [18], which introduces a quadratic term to force the block-diagonal feature for clustering, and it has been proved that SSQP satisfies the block-diagonal feature on the assumption of orthogonal linear subspaces [18]. Following this way, we revise the model in (4) by replacing the low rank constraint of the $Z$ with a a quadratic term to obtain an affinity matrix with the block-diagonal feature,

$$\min_{X,Z,E} \frac{1}{2}\|X - XZ\|_F^2 + \frac{\lambda}{2}\|Z^T Z\|_1 + \lambda_1\|ZS\|_{2,1} + \lambda_2\|X\|_* + \lambda_3\|E\|_1 \tag{5}$$
$$\text{s.t. } W = QX^{\#} + E, Z \geq 0, diag(Z) = 0,$$

where $Z \geq 0, diag(Z) = 0$ is set to get a feasible solution for $Z$ as in [18].

## 4. Solutions

For the problem (4), the algorithm named alternating direction method of multipliers (ADMM) [24] is used to search for the optimized solution. Let $U = ZS$, then problem in Equation (4) is turned to the problem with the augmented Lagrangian with two constraints.

$$\min_{X,Z,U,E} \frac{1}{2}\|X - XZ\|_F^2 + \frac{\lambda}{2}\|Z^TZ\|_1 + \lambda_1\|U\|_{2,1} + \frac{\gamma}{2}\|U - ZR\|_F^2$$
$$+ \lambda_2\|X\|_* + \frac{\lambda_3}{2}\|E\|_1 + \frac{\gamma_1}{2}\|W - RX^\# - E\|_F^2 \tag{6}$$
$$+ \langle F, U - ZR \rangle + \langle G, W - RX^\# - E \rangle$$
$$\text{s.t. } diag(Z) = 0, Z \geq 0,$$

where $F$ and $G$ are Lagrangian multipliers and $\gamma$, $\gamma_1$ are weight parameters for the term $U = ZS$ and $W = RX^\# + E$. Now we can solve Equation (6) by the following four sub-problems for $X$, $Z$, $U$, and $E$ when fixing other variables alternatively.

1. Fix $Z$, $U$ and $E$, solve for $X$ by

$$\min_X f(X) \frac{1}{2}\|X - XZ\|_F^2 + \lambda_2\|X\|_* + \langle G, W - RX^\# - E \rangle + \frac{\gamma_1}{2}\|W - RX^\# - E\|_F^2. \tag{7}$$

equivalently, we have

$$\min_X f(X) = \lambda_2\|X\|_* + \frac{\gamma_1}{2}\|X - (X_k - \frac{\vartheta f(X_k)}{\gamma})\|_F^2. \tag{8}$$

where $\vartheta f(X_k)$ is the derivative of $f(X)$ when $X = X_k$, and $Z = SVD(C)$, $C = X_k - \frac{\vartheta f(X_k)}{\gamma}$.

2. Fix $X$, $U$ and $E$, solve for $Z$ by

$$\min_Z f(Z) = \frac{1}{2}\|X - XZ\|_F^2 + \frac{\lambda}{2}\|Z^TZ\|_1 + \langle F, U - ZS \rangle + \frac{\gamma}{2}\|U - ZS\|_F^2,$$
$$\text{s.t. } diag(Z) = 0, Z \geq 0. \tag{9}$$

To be noted, $Z$ is element-wise nonnegative, so (9) can be turned into:

$$f(Z) = \frac{1}{2}\|X - XZ\|_F^2 + \frac{\lambda}{2}e^TZ^TZe + \langle F, U - ZS \rangle + \frac{\gamma}{2}\|U - ZS\|_F^2, \tag{10}$$

where $e \in R^n$ is a vector of all 1. So the model (9) is a classical convex quadratic problem which has many practical solutions. Here a simple and efficient solution via projected gradient is adopted.

Firstly, we compute the derivative of $f(Z)$ for $Z$,

$$\partial f(Z) = -X^T(X - XZ) + \lambda_1 Ze - FS^T - \gamma(U - ZS)S^T, \tag{11}$$

where $e \in R^{n \times n}$ is a matrix in which every element is 1. Then to obtain the optimized solution, for function $f(Z)$, the derivative is set to be zero and we have

$$XX^TZ + Z(\lambda_1 e + \gamma SS^T) = X^TX + FS^T + \gamma US^T. \tag{12}$$

3. Fix $X$, $E$ and $Z$, and solve for $U$ by

$$\min_U f_1(U) = \lambda_2\|U\|_{2,1} + \langle F, U - ZS \rangle + \frac{\gamma}{2}\|U - ZS\|_F^2. \tag{13}$$

Equivalently, we have

$$\min_U f_1(U) = \lambda_1\|U\|_{2,1} + \frac{\gamma}{2}\|U - (ZS - \frac{1}{\gamma}F)\|_F^2. \tag{14}$$

Denote $Q = ZS - \frac{1}{\gamma}F$, problem (8) can be solved in [14],

$$
U_i = \begin{cases} \frac{\|Q_i\| - \frac{\lambda_1}{\gamma}}{\|Q_i\|} Q_i & \text{if } \|Q_i\| > \frac{\lambda_1}{\gamma} \\ 0 & \text{otherwise,} \end{cases}
\tag{15}
$$

where $U_i$ stands for the $i$-th column of $U$, $Q_i$ stands for the $i$-th column of $Q$ respectively. Here is the closed-form solution.

4. Fix $Z$, $X$ and $U$, solve for $E$ by

$$
\min_E \frac{\lambda_2}{2} \|E\|_1 + \langle G, W - RX^\# - E \rangle + \frac{\gamma_1}{2} \|W - RX^\# - E\|_F^2,
\tag{16}
$$

which is equivalent to

$$
\min_E f_1(E) = \lambda_2 \|E\|_1 + \frac{\gamma}{2} \|E - (W - RX^\# - \frac{G}{\gamma})\|_F^2
\tag{17}
$$

This problem can be easily solved by the current sparse subspace clustering method.

Combining all the above updating formulas, we summarize the step for proposed OSC-NRSfM in Algorithm 1. The parameter setting refers to that in [14]. For Algorithm 1, usually we apply ADMM to find the optimized solution. By solving four sub-problems step by step, we can repeat the searching process to obtain the final solution.

---

**Algorithm 1** Solving data representation of the proposed ordered subspace clustering (OSC)-non-rigid structure-from-motion (NRSfM).

---

**Input:** The input data $X$, maximal iteration number $N$, parameters $\lambda, \lambda_1, \lambda_2, \gamma, \gamma_1$, constant $\rho$.
**Output:** The data representation $Z$.
  1: Initialization: $Z^0 = 0, U^0 = 0, E^0 = 0, X^0 = 0, t = 0, G = 0, F = 0, \rho = 1.3$.
  2: **while** $t < N$ **do**

  3:   Calculate $X^t$ by (7);
  4:   Find $Z^t$ by solving (9);
  5:   Find $U^t$ by solving (13);
  6:   Find $E^t$ by solving (16);
  7:   Update $F \leftarrow F + \gamma(U - ZS)$ ;
  8:   Update $G \leftarrow G + \gamma_1(W - RX^\# - E)$ ;
  9:   Update $\gamma \leftarrow \rho\gamma, \gamma_1 \leftarrow \rho\gamma_1$;
 10:   Update $t \leftarrow t + 1$;
 11: **end while**

---

## 5. Experiments and Analysis

To evaluate performances of state-of-the-art clustering methods together with our proposed method noted by OSC-NRSfM, we conducted experiments on representative datasets. First, we performed face clustering and expression clustering experiments on the BU-4DFE dataset [25]. Second, we widely evaluated our proposed method for complex non-rigid motion using MSR Action3D Dataset. Finally, we tested our method on the Utrecht multi-person motion (UMPM) [26] dataset, which contains 3D joint positions and 2D point tracks of real-world 2D projections out of videos. We compared the proposed OSC-NRSfM method with LRR [14], OSC [16], QOSC [17], CNRMS [10]

and sparse subspace clustering (SSC) [13]. Here we used subspace clustering error (SCE) [16] and normalized mutual information (NMI) to measure clustering results, which are described as follows.

$$SCE = \frac{\text{num.misclssified points}}{\text{total num.of points}},$$ (18)

where the denominator stands for the number of all samples and the numerator stands for the number of misclassified samples.

The parameters $\lambda, \lambda_1, \lambda_2, \lambda_3$ of our methods and the ones of the compared approaches were tuned experimentally according to the experimental results and the parameter setting analysis described in [10,14,16,17].

*5.1. Face Clustering on Dynamic Face Sequence*

The experiments in this section are face clustering with complex conditions on the BU-4DFE dataset [25]. The dataset has 101 subjects (the female/male ratio is 3/2), including different races. Each subject was required to complete six expressions (happy, surprise, sad, angry, fear and disgust). All 2D face images sequence and dynamic 3D face shapes of different subjects were collected simultaneously and some sample images with 2D feature tracks are shown in Figure 1. In this paper, we only used 2D face images sequences to test the proposed method, hence ASM was introduced to get 76 2D feature tracks for each face image. We randomly selected $c = [2, 3, 5, 8, 10, 20, 30]$ subjects out of 101 persons to complete face clustering. For each subject, we took continuous 11 frames for each face expression, and totally 66 frames of 6 expressions are selected for each subject. Therefore, the data matrix for $c$ subjects is $X \in R^{132c \times 76}$. We repeated 30 times of tests, and the $c$ subjects are selected randomly for each test. The optimal parameter settings were $\lambda = 0.01, \lambda_1 = 0.2, \lambda_2 = 0.001, \lambda_3 = 0.05$.

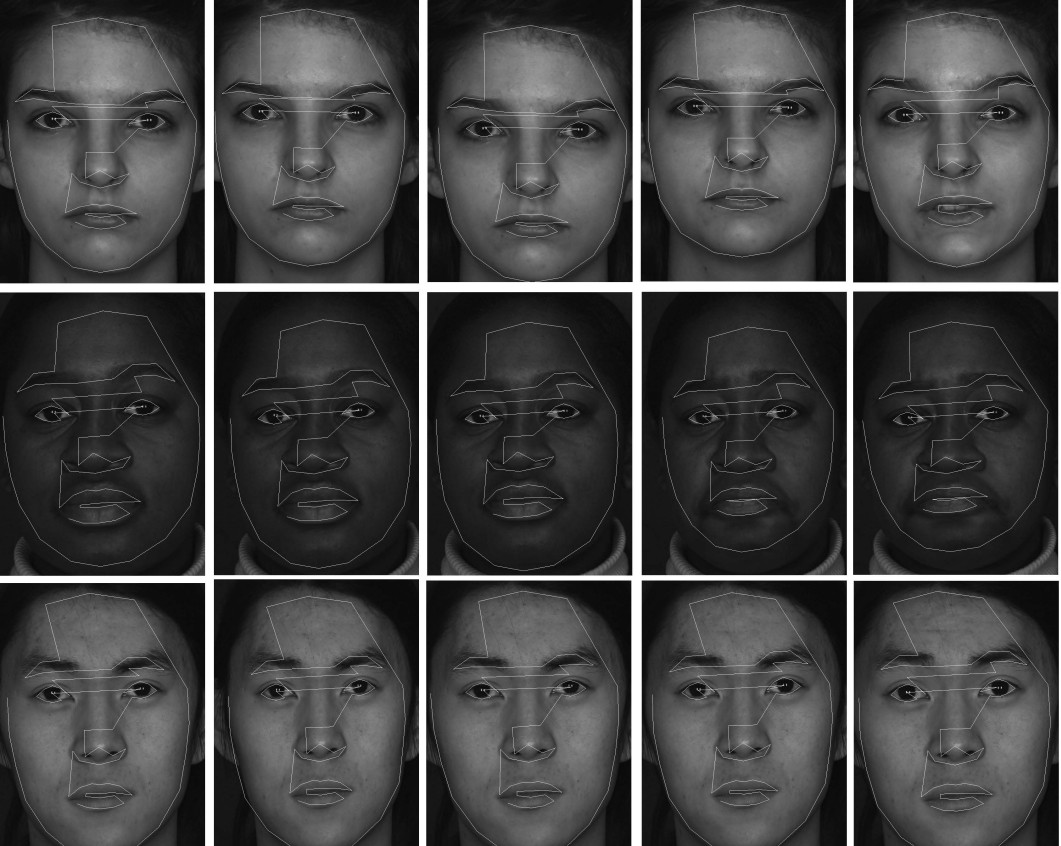

**Figure 1.** Sample images with 2D feature tracks from the BU-4DFE dataset.

It is easy to observe from Table 1 that the proposed method OSC-NRSfM outperformed other methods, except for the case when the number of clusters increased. The results show that the modified methods improved the accuracy of clustering impressively if there were relatively more cluster centers.

**Table 1.** Face clustering error rate (%) on the BU-4DFE dataset for 2,3,5,8,10,20,30 subject classes.

| Subject | The Error Rate(%) | | | | | |
|---|---|---|---|---|---|---|
| | CNRMS | Proposed | LRR | OSC | QOSC | SSC |
| 2 | 14.22 | 6.16 | 34.60 | 0.68 | 0.33 | 47.50 |
| 3 | 19.21 | 1.53 | 35.46 | 1.87 | 0.27 | 63.33 |
| 5 | 24.94 | 0.96 | 27.37 | 3.92 | 2.13 | 66.67 |
| 8 | 28.13 | 0.41 | 28.38 | 5.64 | 4.82 | 78.89 |
| 10 | 26.24 | 0.28 | 26.48 | 8.45 | 7.16 | 78.78 |
| 20 | 30.14 | 2.93 | 34.20 | 9.18 | 6.65 | 80.11 |
| 30 | 34.73 | 5.63 | 36.14 | 10.10 | 7.62 | 82.28 |

*5.2. Expression Clustering on Dynamic 3D Face Expression Sequence*

Facial expressions are very important factors in communications. The BU-4DFE dataset includes 3D face expression sequences for 101 persons, and each person has six expression sequences. Therefore, the expression clustering aims to cluster the face image sequences to 6 categories regardless of the identities. In this experiment, we selected 6 expression sequence from $c$ subjects of the 101 persons to test. For the $i$-th expression sequence of a subject, we picked continuous 11 frames. Then, 2D features of $c$ subjects for $i$-th expression can be represented as $\mathbf{X}_i$, $\mathbf{X}_i (i = 1, 2, \ldots, 6) \in R^{22c \times 76}$. The total face images test dataset is $X = [\mathbf{X}_1, ..., \mathbf{X}_6]$. For one given $c$ from the set $c = [2, 3, 5, 8, 10, 20, 30]$, we randomly selected $c$ subjects to repeat each experiment 30 times and then calculated the average clustering performance as final experimental result. The optimal parameter settings were $\lambda = 0.01, \lambda_1 = 0.02, \lambda_2 = 0.001, \lambda_3 = 0.1$.

The expression clustering error rates on BU-4DFE are shown in Table 2. It can be seen that the performance of proposed OSC-NRSfM method outperforms the compared methods, which demonstrates our proposed method can deal with complex sequences data such as facial expressions sequence. We visualized the difference of affinity matrices of $Z$ when the number of subjects was five in Figure 2. The block-diagonal property of affinity matrices $Z$ demonstrated the block-diagonal constraint to $Z$ is functional. The affinity matrices provided by the proposed method are obviously with block-diagonal features and are more numerous within block weights. The clustering results of $Z$ are shown in Figure 3. Here the same color means the same class it belongs to.

**Table 2.** Expression clustering error (%) on BU-4DFE.

| Subject | The Error Rate(%) | | | | | |
|---|---|---|---|---|---|---|
| | CNRMS | Proposed | LRR | OSC | QOSC | SSC |
| 2 | 31.14 | 15.28 | 49.47 | 28.66 | 19.12 | 52.32 |
| 3 | 32.00 | 16.35 | 60.51 | 21.30 | 20.27 | 67.44 |
| 5 | 45.34 | 20.16 | 66.87 | 26.48 | 21.47 | 77.33 |
| 8 | 50.96 | 12.13 | 70.27 | 32.23 | 24.85 | 79.38 |
| 10 | 55.86 | 8.76 | 72.30 | 32.41 | 23.08 | 79.44 |

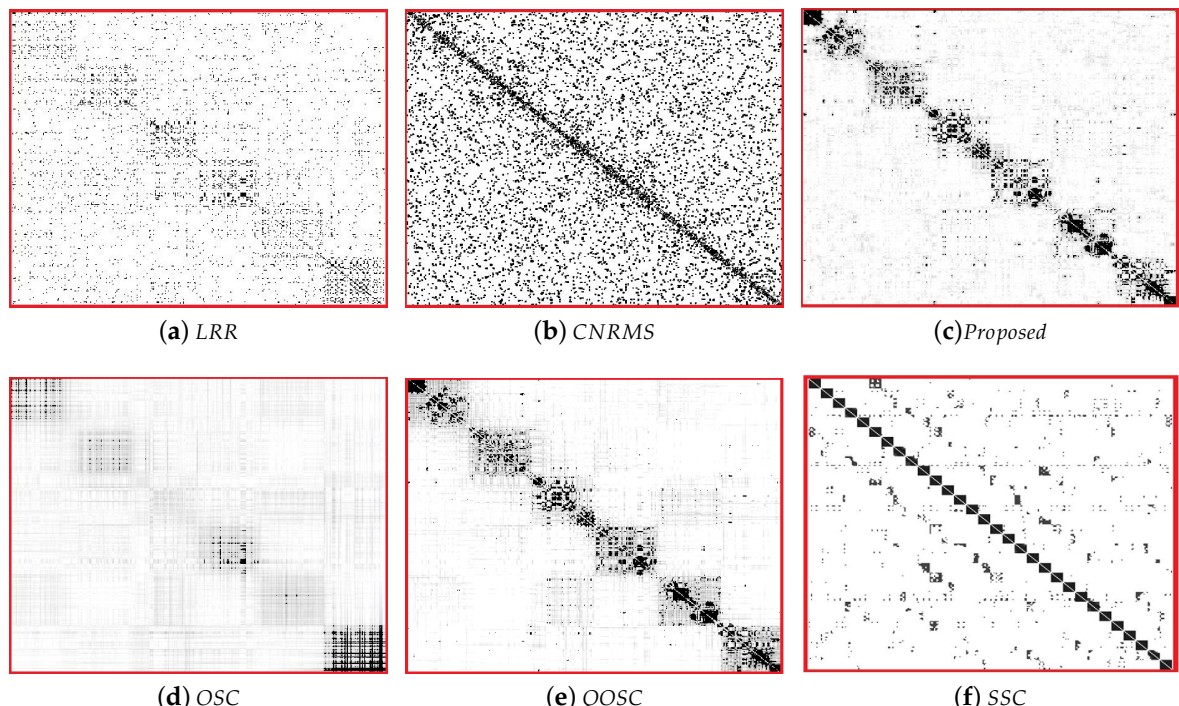

**Figure 2.** The visualization results of affinity matrices Z for expression clustering on the BU-4DFE dataset.

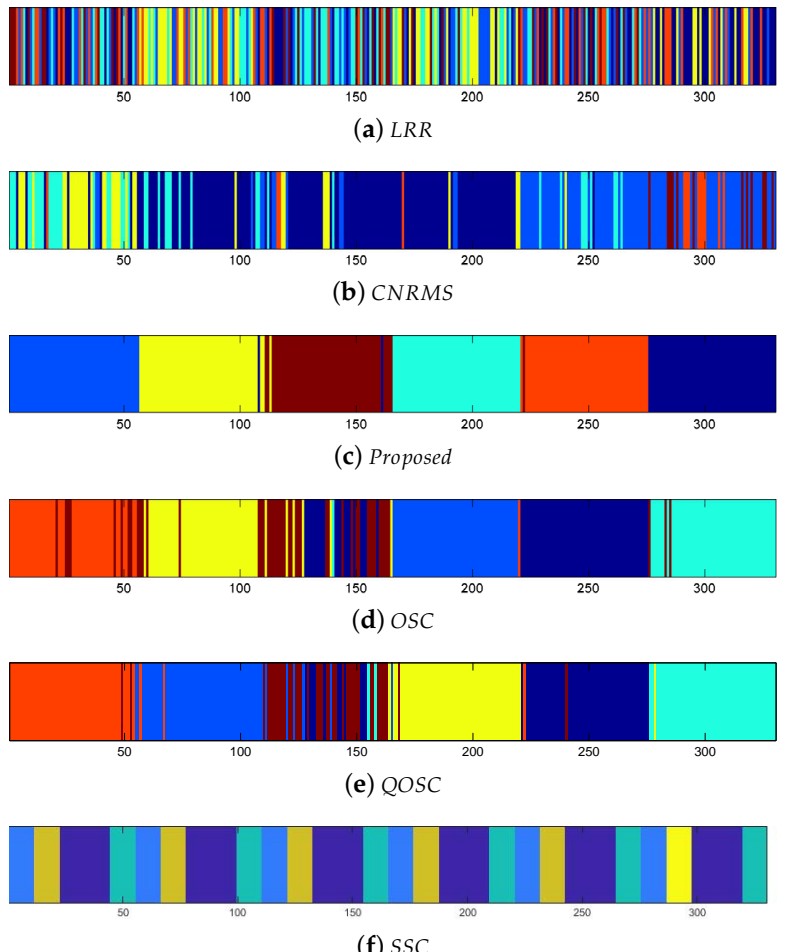

**Figure 3.** Clustering results of six expressions of five subjects on the BU-4DFE dataset.

### 5.3. Clustering on MSR Action3D Dataset

MSR-Action3D is a classical action dataset consist of the depth data. The dataset contains 20 kinds of actions for 10 subjects: high arm wave, hand catch, high throw, horizontal arm wave, two hand wave, hammer, hand clap, draw x, forward kick, draw tick, draw circle, forward punch, sideboxing, jogging, tennis serve, golf swing, tennis swing, bend, side kick, pick up and throw. Each action was performed three times by each subject. The sampling frequency was 15 frames/s, and the spatial resolution of each image was $640 \times 480$. The dataset consists of 23,797 depth maps. Some samples are shown in Figure 4. Many actions were very similar despite clean backgrounds, so the dataset was challenging. To obtain the 2D motion, we utilized a real-time skeleton tracking algorithm [27] to get 20 joint positions in the depth image. In our experiment, we selected $c = [2, 3, 5, 8, 10]$ kinds of actions from 21 kinds of actions randomly. For each selected action, we took continuous 8 frames , therefore, the data matrix is $X \in R^{16c \times 20}$. For one given $c$, we randomly selected $c$ actions to test 30 times so as to calculate the mean clustering errors. The parameter settings are $\lambda = 0.01, \lambda_1 = 0.1, \lambda_2 = 0.001, \lambda_3 = 0.1$. The experimental results are listed in Tables 3 and 4, 3D action clustering error rates and NMI, which demonstrates that the proposed method improved the accuracy of clustering impressively especially when the number of subjects was large. The proposed method was not the fastest one, but it was relatively fast with better performance.

**Table 3.** 3D action clustering error (%) on the MSR Action3D dataset.

| Actions | The Error Rate (%) | | | | | |
|---|---|---|---|---|---|---|
| | CNRMS | Proposed | LRR | OSC | QOSC | SSC |
| 2 | 31.88 | 14.06 | 38.17 | 31.54 | 4.29 | 45.08 |
| 3 | 27.33 | 19.05 | 51.66 | 40.42 | 12.97 | 50.21 |
| 5 | 40.93 | 26.74 | 63.26 | 51.67 | 18.53 | 60.30 |
| 8 | 44.16 | 16.92 | 66.50 | 54.74 | 27.32 | 65.91 |
| 10 | 54.12 | 14.85 | 66.86 | 57.89 | 27.73 | 67.43 |

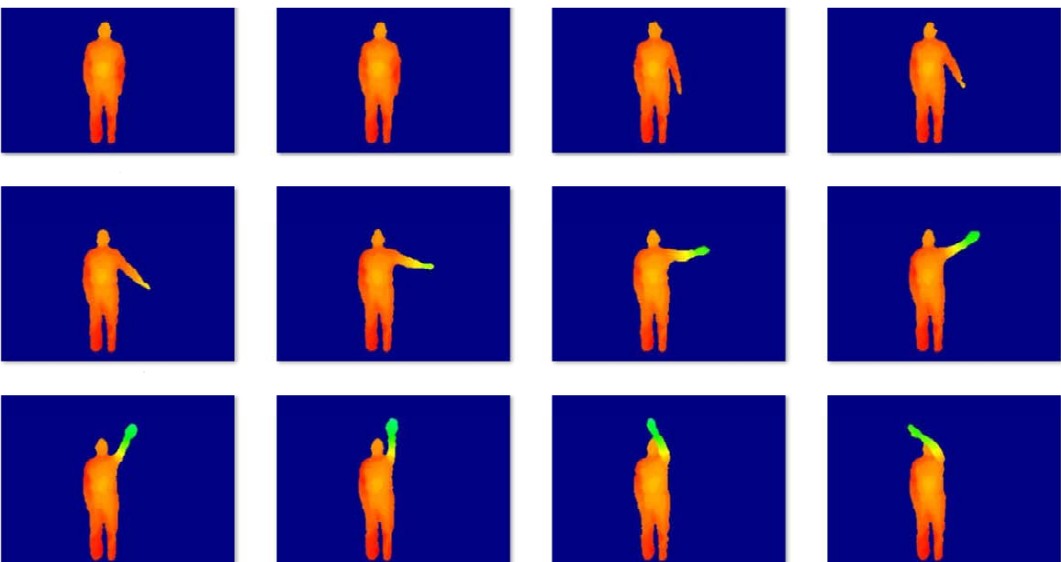

**Figure 4.** Sample images from the MSR Action3D dataset.

**Table 4.** 3D action normalized mutual information (NMI) on MSR Action3D dataset.

| Actions | Normalized Mutual Information (NMI) | | | | | |
|---|---|---|---|---|---|---|
| | CNRMS | Proposed | LRR | OSC | QOSC | SSC |
| 2 | 0.0293 | 0.6674 | 0.1023 | 0.2002 | 0.8157 | 0.0571 |
| 3 | 0.0567 | 0.6425 | 0.1812 | 0.2954 | 0.7910 | 0.2236 |
| 5 | 0.0919 | 0.6606 | 0.2706 | 0.4197 | 0.7044 | 0.3158 |
| 8 | 0.1199 | 0.8348 | 0.3622 | 0.5007 | 0.7335 | 0.3908 |
| 10 | 0.1304 | 0.8428 | 0.4118 | 0.4929 | 0.7504 | 0.4203 |

## 5.4. Clustering on UMPM Motion Dataset

The UMPM Benchmark consists of a set of human motion sequences collected in the real environment, which can be regarded as complex and various non-rigid motion due to several representative daily human actions and interactions of big range change of poses/shapes. Each motion sequences are performed by 1, 2, 3 or 4 persons, and acquired under four fixed viewpoints. The dataset includes the synchronized video recordings and 3D joints information, which were captured by motion capture device. This paper selected eight coherent motion sequences {"$p1 - table - 2$", "$p1 - grab - 3$", "$p1 - chair - 2$", "$p2 - staticsyn - 1$", "$p4 - free - 11$", "$p1 - orthosyn - 1$", "$p3 - ball - 1$", "$p3 - meet - 2$"} to evaluate our method. Here, the video named "$p1 - table - 2$" represents a video recorded by one person performing two actions around the table. The sample images and markers of UMPM dataset [26] are shown in Figure 5.

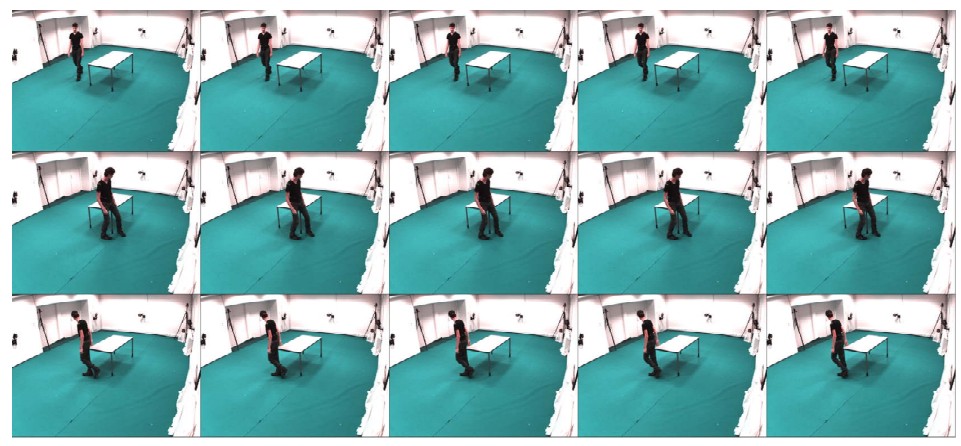

(**a**) Sampled images

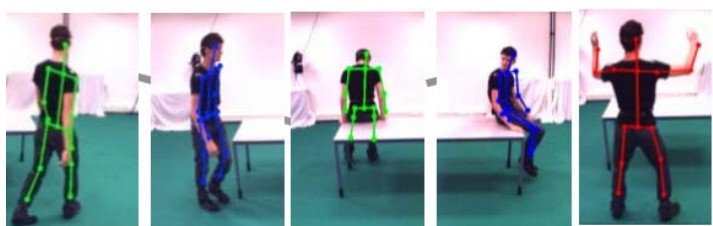

(**b**)Fifteen virtual joint positions per subject

**Figure 5.** Sample images and marker information from the UMPM dataset [26].

As each of the video was relatively large, about 5600 frames, we picked one frame every eight frames, therefore, in total 700 frames were used for testing. We used 15 virtual joint positions per subject as inputs, and then according to the given camera parameters, the corresponding 3D joint position may be reconstructed. In the following, our proposed method will report the ordered subspace clustering result via 3D reconstruction. Since the UMPM dataset contains the known 3D joint positions obtained directly from the motion capture markers. This paper conducts LRR on the 3D ground-truth,

and the other methods are based on the 2D joint coordinates computed from the 3D joint positions. Table 5 shows the quantitative results on the accuracy of clustering. Table 6 shows the NMI results on clustering. In Table 5, the clustering error rate of the proposed method was lower than the other methods, especially, the error of our method was lower than LRR, which was based on the captured 3D joints. This demonstrates that the introduced 3D reconstruction information significantly enhances the accuracy of clustering. Figure 6 is the visual clustering results of $"p3 - meet - 2"$. Here the same color means the same class it belongs to. Small blocks in different colors mean clustering error. Table 7 shows the running time of different methods. The proposed method provides a balance of efficiency and accuracy.

**Table 5.** The clustering error (%) for eight sequences $\{"p1 - table - 2", "p1 - grab - 3", "p1 - chair - 2", "p2 - staticsyn - 1", "p4 - free - 11", "p1 - orthosyn - 1", "p3 - ball - 1", "p3 - meet - 2"\}$.

| Human Motion Sequences | The Error Rate (%) | | | | | |
|---|---|---|---|---|---|---|
| | CNRMS | Proposed | LRR | OSC | QOSC | SSC |
| $p1 - table - 2$ | 31.63 | 25.80 | 36.96 | 33.89 | 30.82 | 33.61 |
| $p1 - grab - 3$ | 18.74 | 15.98 | 25.18 | 20.65 | 19.09 | 54.45 |
| $p1 - chair - 2$ | 22.50 | 13.69 | 27.68 | 22.24 | 21.31 | 14.62 |
| $p2 - staticsyn - 1$ | 3.62 | 2.22 | 6.97 | 5.86 | 3.01 | 16.17 |
| $p4 - free - 11$ | 0.50 | 0.16 | 2.36 | 2.20 | 1.10 | 7.08 |
| $p1 - orthosyn - 1$ | 30.31 | 25.32 | 46.29 | 38.39 | 30.45 | 47.58 |
| $p3 - ball - 1$ | 13.34 | 12.50 | 25.56 | 13.78 | 14.10 | 46.09 |
| $p4 - meet - 2$ | 15.23 | 14.66 | 28.74 | 17.10 | 14.98 | 26.55 |

**Table 6.** The NMI for 8 sequences $\{"p1 - table - 2", "p1 - grab - 3", "p1 - chair - 2", "p2 - staticsyn - 1", "p4 - free - 11", "p1 - orthosyn - 1", "p3 - ball - 1", "p3 - meet - 2"\}$ on UMPM dataset.

| Human Motion Sequences | Normalized Mutual Information (NMI) | | | | | |
|---|---|---|---|---|---|---|
| | CNRMS | Proposed | LRR | OSC | QOSC | SSC |
| $p1 - table - 2$ | 0.4727 | 0.4665 | 0.2926 | 0.4540 | 0.2804 | 0.3638 |
| $p1 - grab - 3$ | 0.3804 | 0.5286 | 0.4639 | 0.4944 | 0.4539 | 0.2403 |
| $p1 - chair - 2$ | 0.4774 | 0.5066 | 0.3514 | 0.3566 | 0.3602 | 0.5904 |
| $p2 - staticsyn - 1$ | 0.7078 | 0.8029 | 0.4546 | 0.4974 | 0.3173 | 0.2518 |
| $p4 - free - 11$ | 0.6411 | 0.9675 | 0.7032 | 0.7684 | 0.7192 | 0.5164 |
| $p1 - orthosyn - 1$ | 0.3139 | 0.3244 | 0.3198 | 0.3202 | 0.2553 | 0.2175 |
| $p3 - ball - 1$ | 0.1842 | 0.3095 | 0.4405 | 0.2871 | 0.3793 | 0.1151 |
| $p4 - meet - 2$ | 0.4619 | 0.6030 | 0.4788 | 0.6049 | 0.5126 | 0.6010 |

**Table 7.** The running time for 8 sequences $\{"p1 - table - 2", "p1 - grab - 3", "p1 - chair - 2", "p2 - staticsyn - 1", "p4 - free - 11", "p1 - orthosyn - 1", "p3 - ball - 1", "p3 - meet - 2"\}$ on UMPM dataset.

| Human Motion Sequences | Running Time (s) | | | | | |
|---|---|---|---|---|---|---|
| | CNRMS | Proposed | LRR | OSC | QOSC | SSC |
| $p1 - table - 2$ | 32,206.98 | 78.20 | 15.03 | 1041.21 | 556.36 | 18.02 |
| $p1 - grab - 3$ | 14,602.75 | 72.20 | 16.41 | 1022.66 | 546.11 | 19.05 |
| $p1 - chair - 2$ | 23,847.55 | 64.41 | 13.75 | 829.50 | 453.08 | 13.58 |
| $p2 - staticsyn - 1$ | 22,581.06 | 620.34 | 198.73 | 784.54 | 424.89 | 7.53 |
| $p4 - free - 11$ | 38,383.29 | 303.11 | 110.44 | 777.50 | 429.48 | 22.50 |
| $p1 - orthosyn - 1$ | 14,616.11 | 75.75 | 88.78 | 739.13 | 396.98 | 7.97 |
| $p3 - ball - 1$ | 50,912.36 | 287.61 | 73.62 | 780.59 | 420.13 | 18.02 |
| $p4 - meet - 2$ | 46,776.58 | 397.76 | 1,204.52 | 943.67 | 509.44 | 22.36 |

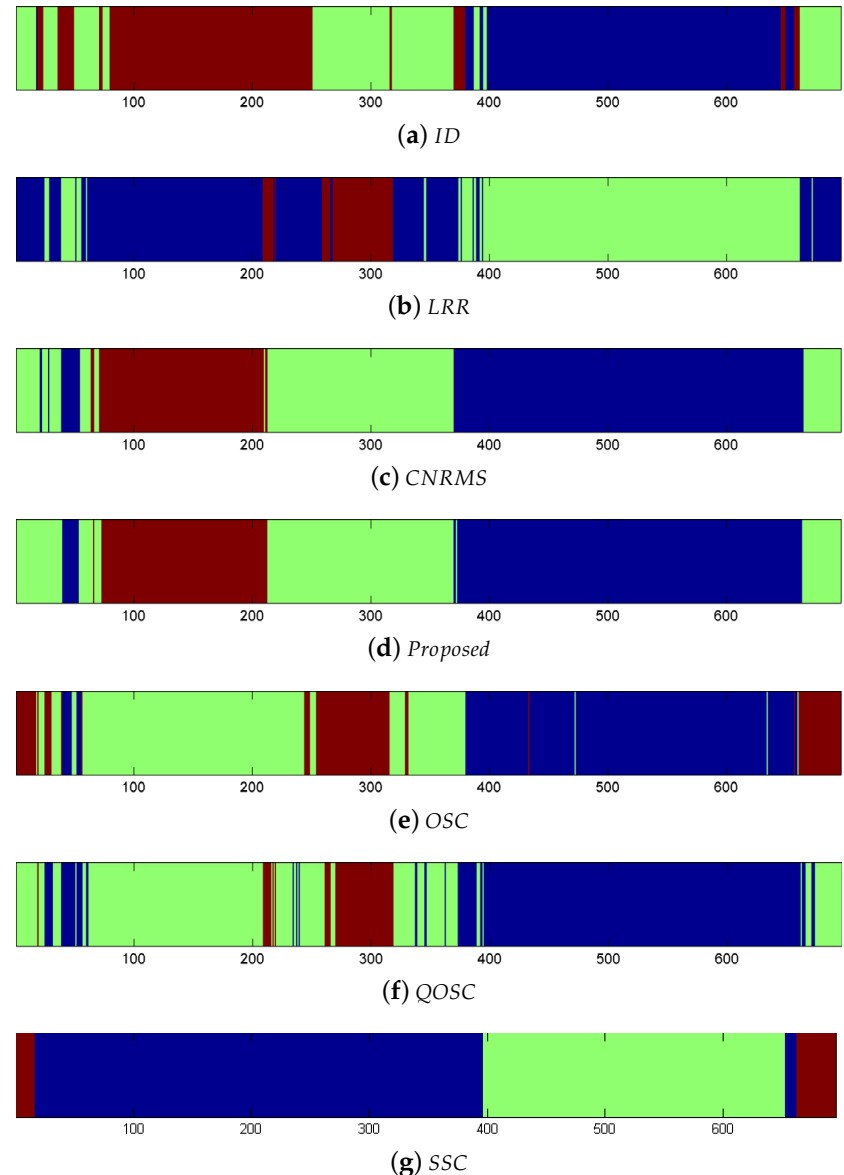

**Figure 6.** Clustering results of "$p3 - meet - 2$".

## 6. Conclusions

The paper has proposed an ordered subspace clustering method for complex and various non-rigid motion via 3D reconstruction. In the proposed model, we reveal the sequential property and intrinsic structure of the complex and various non-rigid motion based on the reconstructed 3D information. Specially, inspired by QOSC and CNRMS, we formulate the block-diagonal structure and sequential constraint to 3D representation generated by CNRMS model so as to obtain good representation for clustering. We verified the proposed method OSC-NRSfM on three public datasets. The experimental results demonstrated that the proposed method of this paper outperforms state-of-the-art methods.

**Author Contributions:** Conceptualization, W.D. and Y.S.; Methodology, J.L.; Software, F.W. and W.D.; Validation, W.D., Y.S. and Y.H.; Formal analysis, J.L.; Investigation, F.W. and W.D.; Resources, Y.H.; Data curation, W.D.; Writing—original draft preparation, W.D., J.L. and F.W.; Writing—review and editing, Y.S., W.D. and Y.H.; Visualization, W.D.; Supervision, Y.S. and Y.H.; Project administration, J.L. and Y.H.; Funding acquisition, Y.S. and Y.H.

**Funding:** This research was funded in part by the National Natural Science Foundation of China under Grant 61876012, 61672066, 61772049, 61602486, in part by the Beijing Educational Committee (KM201710005022).

**Conflicts of Interest:** The authors declare no conflict of interest.

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
