# Peer review of "Ordered Subspace Clustering for Complex Non-Rigid Motion by 3D Reconstruction"

_applsci, doi:10.3390/app9081559_

Round 1
Reviewer 1 Report
In order to take into the account of the inherent sequential property of non-rigid motions, an ordered subspace clustering method is proposed in this study. This paper addressed an important issue in this field. However, the manuscript is not well written and some important information is missing. Please consider my comments to improve the quality of your manuscript.
Abstract:
The recently proposed method to combine NRSfM and subspace representation model is not described clearly, and right after this sentence, you mentioned about the current methods limitations. This section needs to be revised. You do not have to specify one specific method if you are not going to discuss it’s limitations. The abstract is not easily readable to understand the research gap and your proposed framework (or method) and the result achieved. Non-rigid motion analysis is a large topic and you need to specify the subfield which you are going to discuss. How your method outperforms state of the art techniques? Based on which measure? What types of sequential datasets have been used? Mention the name of the datasets (or some of them).
Introduction
Based on the list of your contributions (lines 75), your method is efficient. How do you convince the readers that your method is efficient without comparing the speed with other available tools? This problem should be addressed in the revised version. Are the corresponding approaches based on feature extraction and matching? Are you referring to feature based methods in line 26? You may consider some references for these features and matching methods such as: a new point matching algorithm for non-rigid registration, and, contour-based corner detection and classification by using mean projection transform, and, invariant feature matching for image registration application based on new dissimilarity of spatial features. Please fix the spacing issue in line 60.
Related works
It seems Cand’es method [8] is not directly related to your work. You also mentioned that you are going to discuss [6] and [10], but you started with [8]. What is X in Eq. 1, and why minimizing D since it is clean data? However, it is an introduction to the method [6], is not well written and the connection between different methods are not well described. What type of penalty is lambda in Eq. 3?
You need to discuss OSC and QOSC methods in related works since your method is based on these methods.
Proposed method
Why the penalty term to penalize the similarity between consecutive columns of the low-rank representation Z from reconstructed 3d motion data can help? You need to discuss it.
Solutions
Include more information about each image of Figure 1 in the caption and in the manuscript text before the figure. A detailed discussion and explanation are required in the text for Algorithm 1.
Experiment
You may use more performance metrics rather than just using SCE. A response time comparison is missing to prove the contribution of efficiency you mentioned earlier
You need to discuss each Table and Figure in the text separately. Please include enough information about each one.
Figure 3 is not clearly described in the text. Please provide more information about each section in the Figure caption.
Why Figure 6 is placed in the middle of the references? Please provide enough information in the text.
Author Response
Dear Editor,
Thank you very much for your attention and all the reviewers for their invaluable comments on our manuscript.
We have seriously considered your advice and the reviewers' constructive suggestions and revised the manuscript accordingly. I am submitting the revised version for your further consideration for publication. Attached with the manuscript is a list of the responses to all the reviewers.
Thank you again and look forward to hearing from you soon.
Best regards
Yours sincerely
Jinghua Li

Reviewer 2 Report
@page { margin: 0.79in } p { margin-bottom: 0.1in; line-height: 120% }In equation 1. What represents the letter X?
Somme figures and tables are not linked with the text, for example: figure1, figure 2, table 1 and table 2.
The figure 1 is presented but the results are not discussed or analyzed. Why the figure is presented, why the figure is important?
How were determined the values for the penalty parameters?
In Algorithm 1, the variable rho appears, it was not mention before, what is and why it has a value of 1.3?
In line 150, “the setting described in papers” please cite the papers.
I suggest to show the 2D feature tracks in the figure 2.
From the line 160 to 163, What results are the author talking about, these are not presented or linked.
In line 162, how is determine that the method improve the “reconstruction impressively”. what is the metric used?
In line 166 and 167, avoid to repeat information.
If the subjects are selected randomly, why the subjects used in the expression clustering test are the same used in the face clustering test?
Section 5.3, why are the same subjects used in the last experiments?
In line 187 and 188, how is evaluate the robustness of the method?
In figure 4, the title mention faces but there are actions.
Section 5.4, It is not mention which features were used of the dataset. I suggest to show in the figure 5.
In figure 5, the title mention faces but there are human motions.
Why the OSC-NRSfM was compare against only 4 methods, it is necessary to compare with more methods if it is possible. How is possible to guarantee that others methods not included in the comparison are not better?
The method is named OSC-NRSfM, that means a part of cluster and other of SfM. The theoretical base of the cluster part was presented but the SfM part was not presented.
There are important typos, please fix them in the revision.
Author Response

(The authors gave the same response as above.)

Reviewer 3 Report
This manuscript proposes a novel framework to segment complex and non-rigid motion using ordered subspace clustering by utilizing sequential properties. The research gap is clearly identified and the method is explained with details. Comprehensive experiments are conducted with comparisons to other methods.
However, the novelty of the manuscript is a small improvement on previous methods and the results do not show great improvements. The discussion on experiments is not sufficient. The authors need to analyze the experimental results and discuss the advantages and disadvantages of the proposed approach. In addition, here are some minor grammar errors in the text, please proofread them carefully.
Line 71. for segment --> for segmentation
Line 44: What do you mean by "basic formation"? Please use a better expression.
Line 47. "are popular with good results". Please use more scientific expression and also explain why they can achieve good results theoretically.
L73: Why is using a quadratic constraint is the main contribution of your paper?
Line 94: "data sampled from a subspace like that of RPCA and suffered the same ..." The sentence has grammar errors
Figure 1. some of the pictures are too dark. Could you use colored ones or adjust the brightness of the picture for better understanding?
The authors do make comments on Figure 1. Table 1 and Table 2 are not explicitly referred to in the text.
By the way, which method is the proposed method? The OSC-NRSFM? This should be covered in the experimental section.
figure 4. 5, sample faces images --> sample images
Line 204. Please discuss the advantages and disadvantages of the proposed approach.
Author Response

(The authors gave the same response as above.)

Round 2
Reviewer 1 Report
Thank you for providing the revised version. All the comments have been addressed.
Reviewer 3 Report
All the comments are well addressed.